 eLIFE

# Evolutionary consequences of intra-patient phage predation on microbial populations

**Kimberley D Seed[1], Minmin Yen[1], B Jesse Shapiro[2], Isabelle J Hilaire[3], Richelle C Charles[4,5], Jessica E Teng[3,6], Louise C Ivers[3,6,8], Jacques Boncy[7], Jason B Harris[4,5,9], Andrew Camilli[1]***

[1]Department of Molecular Biology and Microbiology, Howard Hughes Medical Institute, Tufts University School of Medicine, Boston, United States; [2]Département de sciences biologiques, Université de Montréal, Montreal, Canada; [3]Partners In Health, Boston, United States; [4]Division of Infectious Diseases, Massachusetts General Hospital, Boston, United States; [5]Department of Medicine, Harvard Medical School, Boston, United States; [6]Division of Global Health Equity, Brigham and Women's Hospital, Boston, United States; [7]National Public Health Laboratory, Port-au-Prince, Haiti; [8]Department of Global Health and Social Medicine, Harvard Medical School, Boston, United States; [9]Department of Pediatrics, Harvard Medical School, Boston, United States

**Abstract** The impact of phage predation on bacterial pathogens in the context of human disease is not currently appreciated. Here, we show that predatory interactions of a phage with an important environmentally transmitted pathogen, *Vibrio cholerae*, can modulate the evolutionary trajectory of this pathogen during the natural course of infection within individual patients. We analyzed geographically and temporally disparate cholera patient stool samples from Haiti and Bangladesh and found that phage predation can drive the genomic diversity of intra-patient *V. cholerae* populations. Intra-patient phage-sensitive and phage-resistant isolates were isogenic except for mutations conferring phage resistance, and moreover, phage-resistant *V. cholerae* populations were composed of a heterogeneous mix of many unique mutants. We also observed that phage predation can significantly alter the virulence potential of *V. cholerae* shed from cholera patients. We provide the first molecular evidence for predatory phage shaping microbial community structure during the natural course of infection in humans.

*For correspondence: andrew.camilli@tufts.edu

**Competing interests:** The authors declare that no competing interests exist.

**Reviewing editor**: Pascale Cossart, Institut Pasteur, France

## Main Text

Traditional views of host–pathogen interactions are of a battle between two opposing organisms. This perspective is being challenged with greater appreciation of the influence of the host's microbial ecosystem on these interactions (*Lozupone et al., 2012*). Bacteriophages influence bacterial populations in many ecosystems and specifically temperate phages play a role in many diseases through lysogenic conversion (*Brüssow et al., 2004*). In contrast, the impact of lytic phage predation on bacterial pathogens in the context of human disease is under-appreciated. *Vibrio cholerae* is a water-borne pathogen responsible for the diarrheal disease cholera. In order for *V. cholerae* to colonize the human small intestine and elicit diarrhea, it produces a set of virulence determinants including cholera toxin (*Herrington et al., 1988*). The ToxR signaling cascade activates expression of these virulence factors in response to host environmental stimuli (*Miller et al., 1987*; *Taylor et al., 1987*; *DiRita et al., 1991*; *Childers and Klose, 2007*). Cholera epidemics appear with regular seasonality in regions like

**eLife digest** Cholera epidemics occur seasonally in areas such as Bangladesh, and outbreaks can also strike in vulnerable regions, as has occurred recently in Haiti. The disease is caused by *Vibrio cholerae*, a water-borne bacterium that colonizes the small intestine, and its symptoms include severe diarrhea and vomiting which can lead to death if the patient is not treated promptly.

Lytic phages are viruses that specifically attack and kill bacteria. After replicating many times inside the bacterial cell, the phages break open and destroy the cell. Over time a bacterial population can evolve to resist this phage 'predation'; however, it is not known if bacterial pathogens need to defend themselves against phage attack when they infect humans. It had been suggested that phages might affect the progress of cholera infections in people, but molecular evidence that supports this hypothesis was lacking.

When testing stool samples from Haitian cholera patients, Seed et al. found one sample contained a lot of lytic phage relative to the amount of *V. cholerae* present. This phage was very similar to—but distinct from—a phage found in Bangladeshi patients.

The *V. cholerae* bacteria isolated from the stool sample were resistant to attack by the phage. Sequencing the genome of individual bacteria from this sample revealed that each had a mutation that made them resistant to the phage; and while many types of these mutations were found, these were the only differences between all the *V. cholerae* bacteria in this patient sample. This suggests that this resistance developed independently many different times within the patient due to strong selective pressure from phage predation.

When Seed et al. looked at a phage-positive stool sample from a Bangladeshi patient, more mutations that made the bacteria resistant to this phage were found; however, these mutations were different again from the ones in the Haitian bacteria. Because of the nature of these mutations the bacteria from this patient were rendered unable to cause disease and non-transmissible.

This work shows that phages can indeed have access to pathogenic bacteria during human infection. It also indicates that the pressure imposed by phage predation can, in some cases, be so strong that the bacteria lose their virulence and ability to spread to other humans in order to become resistant to the phage.

Bangladesh (*Faruque et al., 2005*) and can arise unpredictably in vulnerable regions as exemplified by the epidemic that recently began in Haiti following the single-source introduction of a pandemic *V. cholerae* O1 strain from another continent (*Chin et al., 2011*; *Cravioto et al., 2011*; *Frerichs et al., 2012*; *Katz et al., 2013*). Lytic phages are hypothesized to impact cholera disease burden in Bangladesh (*Faruque et al., 2005*). The predation of *V. cholerae* by phages following their co-ingestion from the environment is central to this hypothesis; however, molecular evidence in support of this hypothesis is lacking.

We recently described the ICP2 species of *V. cholerae*-specific, virulent podoviruses that are found sporadically in cholera patient stool samples collected since 2001 in Bangladesh (*Seed et al., 2011* and present study). To begin to address the geographic diversity of cholera phages in patient stools, we tested by plaque assay for the presence of phages within nine Haitian cholera patient stool samples collected in 2013. We identified one sample that had a high titer of phage ($10^8$ PFU/ml) relative to *V. cholerae* ($10^5$ CFU/ml). Whole genome sequencing and comparative analysis of this phage, named ICP2_2013_A_Haiti, revealed 84% identity over 93% of its genome to ICP2_2011_A, a phage isolated from Bangladesh in 2011. An alignment of the annotated ICP2 genome (*Seed et al., 2011*) with the three available ICP2-related isolates shows that synteny is conserved between these geographically distinct phages, with no significant genome rearrangements observed (*Figure 1—figure supplement 1*). The Haitian ICP2 isolate is strikingly similar, although clearly distinct, from Bangladeshi ICP2 isolates, and is the first lytic phage reported to be associated with epidemic *V. cholerae* in Haiti.

Remarkably, we observed that most *V. cholerae* isolates recovered from the same stool sample as ICP2_2013_A_Haiti were resistant to infection by this phage. We investigated phenotypic heterogeneity within this stool sample by testing 269 single colony isolates for sensitivity to ICP2_2013_A_Haiti and found that 267 (>99%) were phage-resistant. Eight phage-resistant isolates and the two phage-sensitive isolates from this sample were subjected to whole genome sequencing. All 10 isolates were

isogenic except for mutations within *ompU* encoding the major outer membrane porin (**Supplementary file 1**). Furthermore, the *ompU* mutations in the eight phage-resistant isolates were heterogeneous. We sequenced *ompU* from an additional 11 phage-resistant isolates from this stool sample and found a total of six unique *ompU* alleles (**Figure 1A**). When each of these alleles was used to replace *ompU* in a clean genetic background, all conferred resistance to ICP2, yet produced normal amounts of OmpU (**Figure 2** and data not shown), suggesting that ICP2 uses OmpU as a receptor to initiate infection and that the mutations disrupt this interaction. These data showing that intra-host *V. cholerae* isolates are isogenic, only differing in ICP2 resistance mutations, indicates that they diverged within the patient. Furthermore, the presence of multiple different ICP2 resistance mutations within this single host suggests that selection of phage resistance occurred multiple, independent times during the course of infection. We also found evidence for ICP2-mediated selection of OmpU mutants in isolates from Bangladesh. Analysis of the *ompU* sequence from 54 clinical isolates collected between 2001 and

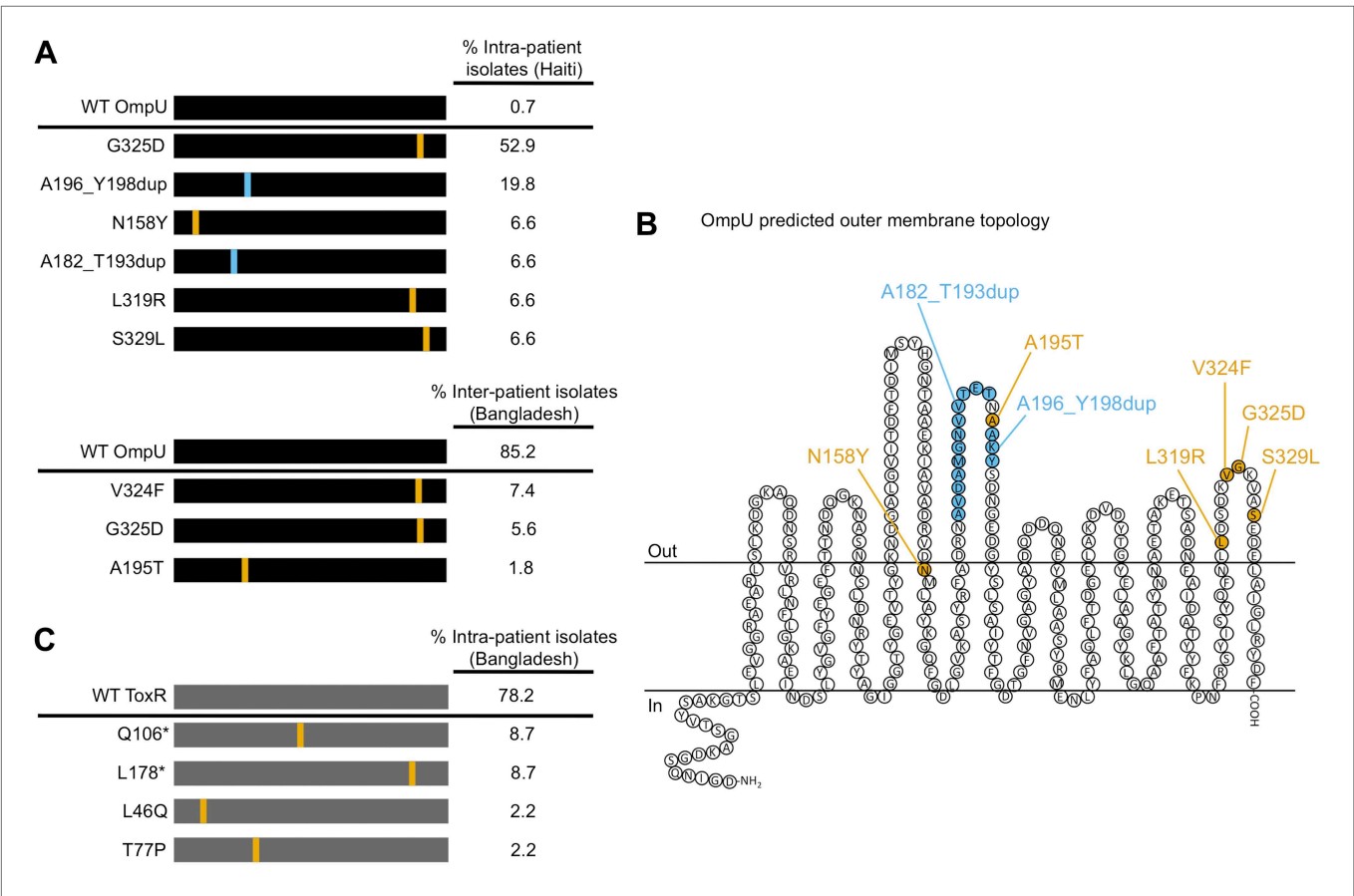

**Figure 1**. The presence of *V. cholerae* OmpU and ToxR mutants present within and between cholera patients. (**A**) Graphical depiction and frequency of OmpU mutants found within a stool sample containing ICP2_2013_A_Haiti phage ($10^8$ PFU/ml) from a single Haitian patient (top) and from different patients in Bangladesh (n = 54) (bottom). (**B**) Predicted membrane topology of mature OmpU generated using Pred-TMBB (**Bagos et al., 2004**). Locations of amino acid substitutions or insertions carried by *V. cholerae* clinical isolates are indicated. (**C**) Graphical depiction and frequency of ToxR mutants found within a stool sample containing ICP2_2011_A ($10^9$ PFU/ml) from a single Bangladeshi patient. Amino acid substitutions or nonsense mutations (asterisks) are in orange and duplications are in blue.

The following figure supplements are available for figure 1:

**Figure supplement 1**. ICP2_2013_A_Haiti is closely related to ICP2 bacteriophages from Bangladesh.

**Figure supplement 2**. Identification of OmpU mutants in samples collected at the International Centre for Diarrheal Disease Research, Bangladesh between 2001–2011.

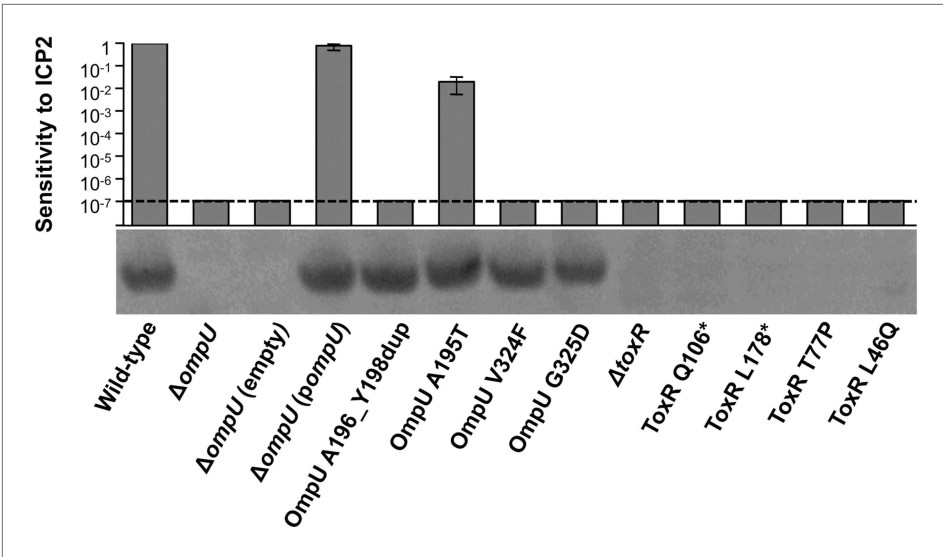

**Figure 2**. OmpU expression of OmpU and ToxR mutants and their sensitivity to ICP2. Outer membrane fractions were prepared from samples matched by equivalent $OD_{600}$ units. Samples were separated by SDS-PAGE and subjected to Western blot analysis using rabbit polyclonal antisera against OmpU. The sensitivity of each strain to ICP2_2013_A_Haiti is represented as a histogram of the efficiency of plaquing, which is the plaque count ratio of a mutant *V. cholerae* strain to that of the wild-type strain. The limit of detection for plaque assays was $10^{-7}$.

2011 showed that 15% had non-synonomous mutations (**Figure 1A**, **Figure 1—figure supplement 2**) and these alleles were sufficient for ICP2 resistance (**Figure 2**). One of the OmpU mutants, G325D, was observed in clinical isolates from Haiti and Bangladesh (**Figure 1A**). Interestingly, seven of the eight mutant OmpU proteins detected in the Haitian and Bangladeshi clinical isolates had alterations in two predicted extracellular loops, while the eighth had an altered transmembrane segment adjacent to the first of these extracellular loops (**Figure 1B**). This suggests that ICP2 interacts specifically with these exposed loops to initiate infection.

We also tested for *V. cholerae* population heterogeneity within an ICP2-positive stool sample isolated in Bangladesh in 2011. Of 46 single colony isolates tested for sensitivity to ICP2_2011_A, 22% were resistant to the phage. Whole genome sequencing of two phage-sensitive and four phage-resistant isolates from this sample revealed that the strains were isogenic, except that phage-resistant isolates had nonsense mutations in *toxR*, the direct transcriptional activator of *ompU* (**Crawford et al., 1998**; **Supplementary file 2**). We sequenced *toxR* from an additional eight phage-resistant isolates from this stool sample and found that, analogous to what was observed in the Haitian patient sample, there were multiple unique *toxR* alleles, which included two non-synonomous and two nonsense mutations (**Figure 1C**). Each of the four mutant *toxR* alleles conferred ICP2 resistance and abrogated expression of OmpU when used to replace wild-type *toxR* in a clean genetic background (**Figure 2**). Both ICP2-sensitivity and OmpU expression were restored to the clinical *toxR* mutants by expressing *ompU in trans* or by reverting the *toxR* allele to wild-type (data not shown), indicating that these mutant *toxR* alleles are necessary and sufficient for ICP2 resistance and that this resistance is mediated through loss of OmpU expression.

To address the potential consequences of the phage-resistance mutations on *V. cholerae* fitness, we tested the four most frequently isolated *ompU* alleles and all four *toxR* alleles in survival and growth assays. Recent Tn-seq analyses of *V. cholerae* showed that OmpU is critical for fitness in an infant rabbit infection model (**Fu et al., 2013**; **Kamp et al., 2013**) and for dissemination from the host into pond water (**Kamp et al., 2013**). OmpU has also been shown to be important for protection against the bactericidal effect of bile salts (**Provenzano et al., 2001**), cationic peptides (**Mathur and Waldor, 2004**), and intestinal organic acids (**Merrell et al., 2001**). When we tested for survival in bile and for competitive fitness in pond water we found that the four OmpU mutants are fully fit (**Figure 3A,B**),

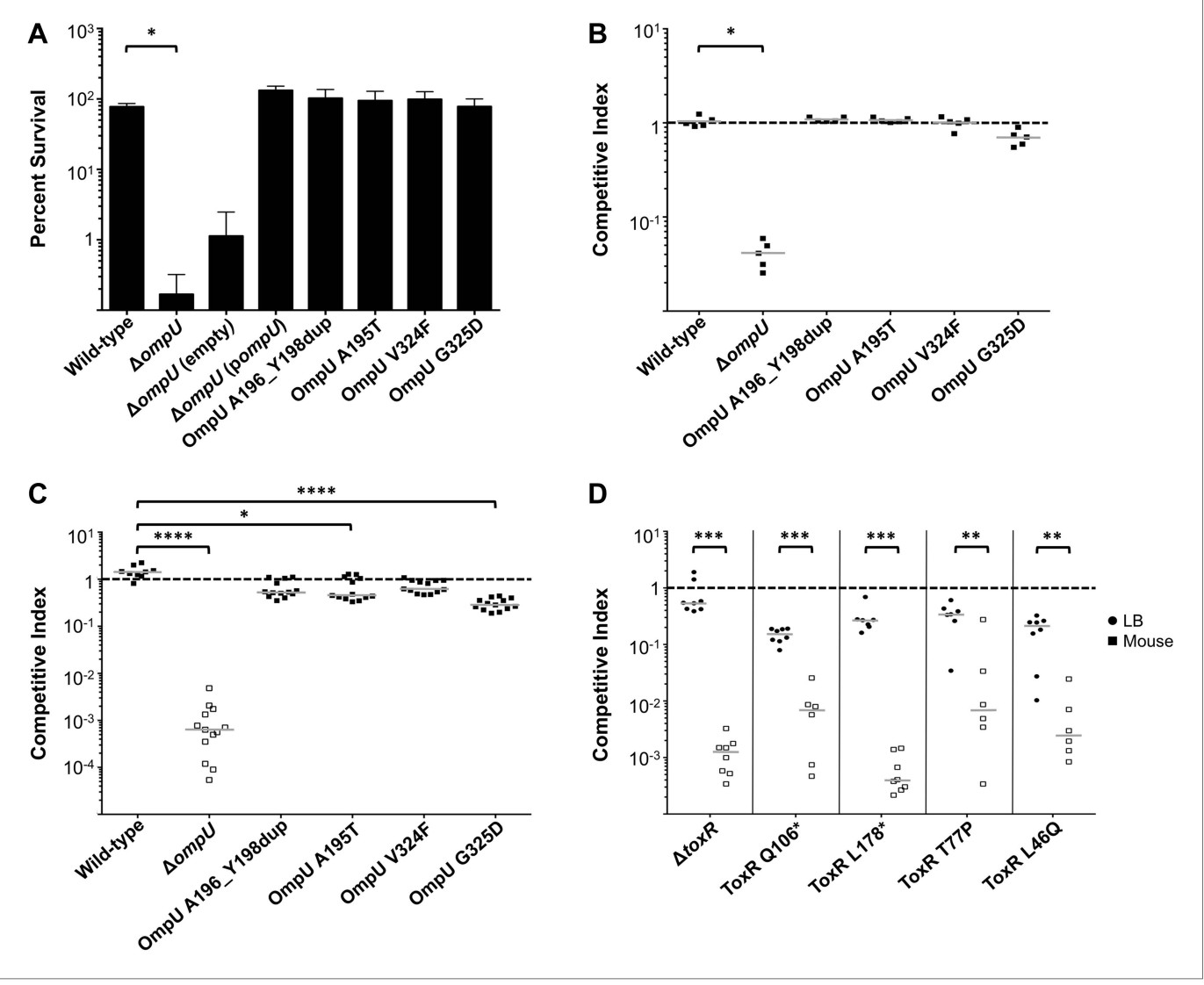

**Figure 3**. The fitness cost of clinically relevant OmpU and ToxR mutations. (**A**) Clinically relevant OmpU mutants retain fitness in the presence of bile. *p < 0.05 significantly different means for the compared data sets (Mann–Whitney U Test). (**B**) OmpU mutants retain competitive fitness in pond water. *p < 0.05 significantly different from wild-type control (Kruskal–Wallis and *post hoc* Dunn's multiple comparison tests). (**C**) OmpU mutants have slight competitive fitness defects when serially passaged in Luria–Bertani broth (for ca. 58 generations). *p < 0.05 or ****p < 0.0001 significantly different from wild-type control (Kruskal–Wallis and *post hoc* Dunn's multiple comparison tests). (**D**) ToxR mutants are attenuated in vivo using the infant mouse colonization model. **p < 0.01 or ***p < 0.001 significantly different from the in vitro median (Mann–Whitney U Tests). The horizontal bars indicate the median of each data set. Open symbols represent data below the limit of detection.

The following figure supplement is available for figure 3:

**Figure supplement 1**. Clinical isolates harboring ToxR mutations are severely attenuated for infection.

consistent with their wild-type levels of OmpU expression (*Figure 2*). Moreover, the OmpU(G325D) mutant was fully virulent in a single round, competitive infection in infant rabbits (*Figure 4*, first column). However, the OmpU mutants, particularly the A195T and G325D mutants, showed a mild competitive defect after multiple passaging in growth medium (*Figure 3C*). The mild growth defect of the OmpU mutants in conjunction with the low prevalence of ICP2 (*Seed et al., 2011*) may explain why the ICP2-resistant *ompU* variants do not become fixed in the population (*Figure 1—figure supplement 2*). In sharp contrast, in the presence of ICP2 we observed a 10,000-fold enrichment of the OmpU(G325D)

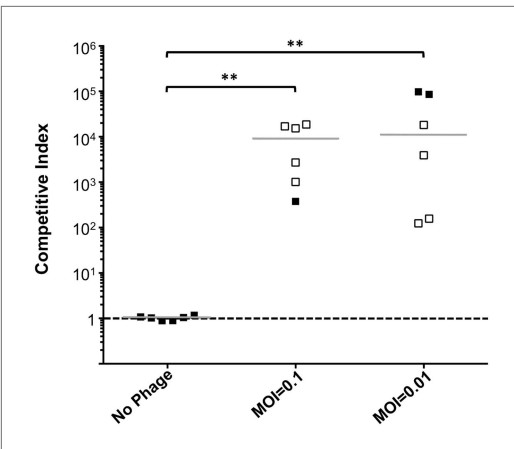

**Figure 4.** Phage predation leads to enrichment of OmpU mutant over wild-type in vivo. Competitive indices (CI) were determined between wild-type ΔlacZ and OmpU G325D in the absence or presence of ICP2_2013_A_Haiti at the multiplicity of infection (MOI) indicated in infant rabbits 12 hr post-infection. Each symbol represents the CI for an individual rabbit and the horizontal lines indicate the median for each condition. The open symbols represent data below the limit of detection for the wild-type strain. **p < 0.01, significantly different from no phage control (Kruskal–Wallis and *post hoc* Dunn's multiple comparison tests). DOI: 10.7554/eLife.03497.009

mutant over the wild-type after infection of infant rabbits (*Figure 4*), indicating that strong selective pressure is imposed by phage predation during *V. cholerae* infection. This emulates what we hypothesize happens in human infections in the presence of ICP2 and is the first demonstration of phage predation of *V. cholerae* in the context of a diarrheal disease model.

*V. cholerae* strains harboring any of the four ToxR mutations were avirulent in the infant mouse model of infection and were indistinguishable from a Δ*toxR* mutant (*Figure 3D*). The clinical isolates containing these mutations were also at least 100-fold attenuated (*Figure 3D—figure supplement 1*), indicating that these isolates do not harbor compensatory mutations. The cytoplasmic amino terminus (D17-E112) of ToxR where the two non-synonymous mutations mapped has homology to the winged helix-turn-helix family of transcription activators and is involved in DNA binding and transcriptional activation (*Miller et al., 1987*; *Ottemann et al., 1992*). A number of point mutations in this domain were previously shown to abrogate activation of *ompU* and virulence genes (*Ottemann et al., 1992*; *Morgan et al., 2011*). The severe virulence defects of the two non-synonymous *toxR* mutants from the Bangladeshi patient sample (*Figure 3D*) are consistent with an inability of the encoded mutant ToxR proteins to activate these genes. These *toxR* mutants would not colonize the human small intestine and be recovered in the secretory diarrhea if ingested (*Herrington et al., 1988*), which further supports our conclusion that these mutants arose due to ICP2 predation during cholera infection in this individual. This also indicates that, in this patient, predation and selection of phage-resistant mutants likely occurred late in infection when colonization and virulence gene expression were no longer required for the development of acute symptoms (*Merrell et al., 2002*).

We have shown that lytic phages are an unexpected 'third party' that impose significant bactericidal pressure on an environmentally transmitted pathogen during the natural course of infection in humans. We observed that adaptations to phage predation involve tradeoffs in evolutionary fitness and provide a molecular mechanism for phage predation impacting *V. cholerae* transmission and seeding of environmental reservoirs. Our results highlight that host–pathogen interactions are embedded within and strongly affected by a complex microbial ecosystem. Predator-prey dynamics are notably absent between microbiotal phage and their bacterial hosts in the intestinal ecosystem in healthy humans (*Reyes et al., 2010*). In contrast, our findings indicate that diseased states, which are often accompanied by significant bacterial proliferation, likely facilitate these predatory interactions.

## Materials and methods

### Growth conditions, strains and genomic analysis

Strains utilized in this study are listed in *Supplementary file 3*. Stool specimens and *V. cholerae* isolates from the International Centre for Diarrheal Disease Research, Bangladesh were collected and stored from previous studies (*Nelson et al., 2008*; *Seed et al., 2011*, *2012*, *2013*). Phage (ICP2_2011_A and ICP2_2013_A_Haiti) were isolated from cholera rice-water stool samples as described (*Seed et al., 2011*). *V. cholerae* isolates were isolated from stool samples on Luria–Bertani agar with sulfamethoxazole (24 µg/ml), trimethoprim (32 µg/ml), nalidixic acid (20 µg/ml) and streptomycin (Sm) (100 µg/ml).

Mutations were introduced using pCVD442-lac as previously described (*Seed et al., 2012*). Strains containing the pMMB67EH vector were grown in the presence of 50 µg/ml ampicillin and 1 mM iso-propyl-ß-D-thiogalactopyranoside (IPTG).

*V. cholerae* and phage genomic libraries were generated (*Lazinski and Camilli, 2013*) and sequenced using an Illumina HiSeq2000 (Tufts University Core Facility). Sequence reads from each *V. cholerae* isolate were mapped to the reference genome *V. cholerae* O1 2010EL-1786 (for Haitian isolates, accession numbers NC_016445.1 and NC_016446.1) or to *V. cholerae* O1 N16961 (for Bangladeshi isolates, accession numbers NC_002505.1 and NC_002506.1) and mutations were identified using Varscan v2.3.6 (*Koboldt et al., 2012*) and CLC Genomic Workbench software (Version 6.8; CLC Bio, Denmark). Variants were called as being present (different allele than reference), absent (the same allele as reference) or low-coverage (<15×; in which case no call was made).

## Preparation of outer membrane fractions and Western blotting

Cultures were grown to mid-exponential growth phase in Luria–Bertani (LB) broth at 37°C with aeration. Bacteria were re-suspended in 200 mM Tris–HCl, pH 8.0 and 2 mM EDTA. Sucrose was added to a final concentration of 20% followed by lysozyme treatment for 10 min at 37°C. Using a dry ice/ethanol bath, samples were freeze-thawed twice followed by DNAse I treament for 20 min at room temperature. Samples were spun for 5 min at 16,100×$g$ to pellet the membrane fraction. The pellets were re-suspended in 1% Triton X-100, 10 mM MgCl$_2$, and 50 mM Tris–HCl, pH 8.0 and incubated at 37°C for 20 min. Samples were spun again for 5 min at 16,100×$g$ to separate the inner membrane from the outer membrane. The pellets, which contain the outer membrane proteins, were re-suspended in 200 mM Tris–HCl, pH 8.0.

Outer membrane fractions were boiled for 10 min in sample buffer containing sodium dodecyl sulfate and β-mercaptoethanol and separated on a NuPAGE 4–12% Bis-Tris polyacrylamide gel (Life Technologies, Carlsbad, CA). Protein gels were transferred to a nitrocellulose membrane for Western blotting. Membranes were probed with rabbit polyclonal antisera against *V. cholerae* OmpU (gift of James Kaper). A Cy5 goat anti-rabbit antibody was used to develop the blot.

## Evaluation of in vitro fitness

Strains (grown to OD$_{600}$ = 0.5) were assessed for their ability to survive 0.2% porcine bile (Sigma, St. Louis, MO) in 0.85% NaCl for 1 hr at room temperature. Pond water survival assays were done as described previously (*Kamp et al., 2013*) at 30°C for 48 hr. Competition experiments were performed between the strain of interest (*lacZ*⁺) and the appropriate control strain (Δ*lacZ*), and outputs were plated on LB agar plates containing 100 µg/ml Sm and 40 µg/ml 5-bromo-4-chloro-3-indolyl-β-D-galactopyranoside (X-gal).

## Animals

All animal experiments were in accordance with the rules of the Department of Laboratory Animal Medicine at Tufts Medical Center. Infant mouse colonization assays and in vitro controls were done as described previously (*Seed et al., 2012*). 3-day old infant rabbits were pre-treated with Cimetidine-HCL (Morton Grove Pharmaceuticals, Morton Grove, IL) 3 hr prior to infection (*Kamp et al., 2013*) and infected with ~5 × 10⁸ CFU in 2.5% sodium bicarbonate buffer (pH 9). A 1:1 mixture of wild-type to mutant (for experiments without phage) or 10:1 wild-type to mutant (for experiments including phage) was used. For experiments involving the addition of phage, phage were added immediately before intragastric inoculation to each rabbit inoculum to limit phage adsorption ex vivo (phage were in contact with the bacterial inoculum for ~30–60 s prior to gavage of each animal). Rabbits were euthanized ~12 hr post-inoculation and cecal and/or small intestinal fluid was collected by puncture. As with the in vitro assays, the competing strains were enumerated on LB agar plates containing 100 µg/ml Sm and 40 µg/ml X-gal.

## Acknowledgements

The authors thank the Tufts University Core Facility for sequencing and computational support, James Kaper for providing OmpU antisera, and David Lazinski and Linc Sonenshein for commenting on the manuscript. This work was supported by US National Institutes of Health grants AI055058 (AC), AI099243 (JH/LI), AI089721 (RCC), a Massachusetts General Hospital Physician Scientist Development

Award (RCC), the Natural Sciences and Engineering Research Council of Canada and the Canada Research Chairs program (BJS), and the Charles A King Trust Postdoctoral Fellowship Program (KDS). AC is a Howard Hughes Medical Institute Investigator.

## Additional information

### Funding

| Funder | Grant reference number | Author |
|---|---|---|
| Howard Hughes Medical Institute | | Andrew Camilli |
| National Institutes of Health | AI055058 | Andrew Camilli |
| Massachusetts General Hospital | | Richelle C Charles |
| Natural Sciences and Engineering Research Council of Canada | 950-228996 | B Jesse Shapiro |
| Canada Research Chairs | 950-228996 | B Jesse Shapiro |
| Charles A. King Trust | | Kimberley D Seed |
| National Institutes of Health | AI099243 | Louise C Ivers, Jason B Harris |
| National Institutes of Health | AI089721 | Richelle C Charles |

The funders had no role in study design, data collection and interpretation, or the decision to submit the work for publication.

### Author contributions

KDS, MY, AC, Conception and design, Acquisition of data, Analysis and interpretation of data, Drafting or revising the article; BJS, Analysis and interpretation of data, Drafting or revising the article; IJH, RCC, JET, Drafting or revising the article, Contributed unpublished essential data or reagents; LCI, JB, Conception and design, Drafting or revising the article, Contributed unpublished essential data or reagents; JBH, Conception and design, Analysis and interpretation of data, Drafting or revising the article, Contributed unpublished essential data or reagents

### Ethics

Human subjects: The studies were approved by the institutional review boards of Partners HealthCare (Brigham and Women's Hospital and Massachusetts General Hospital) and Tufts University School of Medicine, and the Haitian National Ethics Committee. Written informed consent was obtained from adult participants and from guardians of children.
Animal experimentation: This study was performed in strict accordance with the recommendations in the Guide for the Care and Use of Laboratory Animals of the National Institutes of Health. All of the animals were handled according to approved institutional animal care and use committee (IACUC) protocol (#B2013-44) of Tufts University School of Medicine.

## Additional files

### Supplementary files

• Supplementary file 1. *V. cholerae* isolates recovered from the ICP2-positive Haitian patient sample are isogenic except for mutations in *ompU*.

• Supplementary file 2. *V. cholerae* isolates recovered from the ICP2-positive Bangladeshi patient sample are isogenic except for mutations in *toxR*.

• Supplementary file 3. Strains used in this study.

## Major datasets

The following datasets were generated:

| Author(s) | Year | Dataset title | Dataset ID and/or URL | Database, license, and accessibility information |
|---|---|---|---|---|
| Seed KD, Yen M, Shapiro BJ, Hilaire IJ, Charles RC, Teng JE, Ivers LC, Boncy J, Harris JB, Camilli A | 2014 | Sequence for ICP2_2011_A | http://www.ncbi.nlm.nih.gov/nuccore/?term=KM224878 | Publicly available at NCBI GenBank. |
| Seed KD, Yen M, Shapiro BJ, Hilaire IJ, Charles RC, Teng JE, Ivers LC, Boncy J, Harris JB, Camilli A | 2014 | Sequence for ICP2_2013_A_Haiti | http://www.ncbi.nlm.nih.gov/nuccore/?term=KM224879 | Publicly available at NCBI GenBank. |
| Seed KD, Yen M, Shapiro BJ, Hilaire IJ, Charles RC, Teng JE, Ivers LC, Boncy J, Harris JB, Camilli A | 2014 | Sequences for the Haitian and Bangladeshi V. cholerae isolates | http://www.ncbi.nlm.nih.gov/sra/?term=PRJNA255987 | Publicly available at NCBI Sequence Read Archive. |

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
