## [Decision Letter]

Thank you for sending your work entitled “Evolutionary Consequences of Intra-Patient Phage Predation on Microbial Populations” for consideration at *eLife*. Your article has been favorably evaluated by Richard Losick (Senior editor), a Reviewing editor and 2 reviewers, one of whom, Marie-Agnès Petit, has agreed to reveal her identity.

The Reviewing editor and the two reviewers discussed their comments before we reached this decision, and the Reviewing editor has assembled the following comments to help you prepare a revised submission.

Seed et al. report a very clear cut and elegant set of data providing evidence of a natural, *in vivo* predator prey relationship between *Vibrio cholerae* and ICP2-like virulent phages of the podoviridae family, upon human infection. In two patients infected with *V. cholerae*, and having high loads of ICP2 phage, they discovered two different panels of Vibrio mutations, in ompU and toxR, resulting in the mutation or repression of the OmpU receptor, respectively. They also establish that in a collection of 54 isolates of *V. cholerae*, 15% had an ompU mutation.

To our knowledge, this is the first report of a predator-prey evolution taking place in the human intestine. In addition, the toxR mutants have attenuated virulence, suggesting that the presence of the virulent phage may have contributed to the elimination of the infection by natural evolution. This is not true however for the ompU mutants that appear almost not attenuated in virulence/ survival phenotypes. The contrast between the two sets of mutants isolated is striking. As suggested by the authors, such an evolution, not reported until now for bacterial strains in healthy microbiota, may have taken place due to the disequilibrium of the resident microbiota in patients. This is a new concept, and the study certainly deserves publication.

This study also represents an extremely interesting and timely topic in light of the recent phage therapy re-birth. One of the reviewers wrote: “I was amazed by the results on ompU mutant selection by phage co-infection in these conditions and was surprised that this selective effect had never been observed in the cholerae OmpU. I went to Genbank and retrieved the various cholerae OmpU (VC0633) sequences and compared its variation to its neighboring D-alanyl-D-alanine carboxypeptidase (VC0632). I did not perform a statistical analysis, but it is obvious that OmpU is more variable than VC0632. However, this can have many different causes, but strikingly, most of the OmpU variants carry mutation(s) in the very same positions as those identified in this study (A182, VG 324/5...). A simple but thorough analysis of the OmpU mutations could add more weight to the conclusions presented in this manuscript on the selective force exerted by phage over the time.”

---

## [Author Response]

[…] This study also represents an extremely interesting and timely topic in light of the recent phage therapy re-birth. One of the reviewers wrote: “I was amazed by the results on ompU mutant selection by phage co-infection in these conditions and was surprised that this selective effect had never been observed in the cholerae OmpU. I went to Genbank and retrieved the various cholerae OmpU (VC0633) sequences and compared its variation to its neighboring D-alanyl-D-alanine carboxypeptidase (VC0632). I did not perform a statistical analysis, but it is obvious that OmpU is more variable than VC0632. However, this can have many different causes, but strikingly, most of the OmpU variants carry mutation(s) in the very same positions as those identified in this study (A182, VG 324/5...). A simple but thorough analysis of the OmpU mutations could add more weight to the conclusions presented in this manuscript on the selective force exerted by phage over the time.”

We thank the reviewers for their comments and idea to include more OmpU sequences. We used BLASTP to identify OmpU in other *Vibrio cholerae* isolates in GenBank. Homologues were aligned and WebLogo (Crooks *et al.*, 2004) was used to generate a sequence logo in order to gain insight into variable or conserved regions. We have included this analysis in a figure below (Figure 5).Author response image 1.

We have also included a reference image equal to Figure 1 but with numbering of the extracellular loops to orient the reader. The mutations observed in *V. cholerae* isolates in this study, and experimentally validated to confer ICP2 resistance, are highlighted in yellow. The results of this analysis indicate that there is significant variability in OmpU, specifically in the predicted extracellular loops. As the reviewers point out, there are other *V. cholerae* isolates in GenBank that share the mutations identified in this study. There are also highly variable loops (i.e., loops 2 and 6) that we did not observe mutations in our data set, and it is not known if variability in these loops can confer phage resistance.

The strains included in the analysis below represent a broad collection of both environmental and clinical *V. cholerae* isolates from geographically disparate locations. Trying to identify the source of these deposited sequences is not always possible. Since OmpU represents an abundant outer membrane protein it is not surprising to observe this variability, which as the reviewers point out, is likely due to many different causes. One could imagine that the variability may be due to immune pressure, or phage pressure from ICP2 and/or other uncharacterized phages. To-date ICP2 like phages have only been isolated from clinical samples in Haiti and Bangladesh and we do not know if they co-exist with environmental non-toxigenic *V. cholerae* in these locations, or for that matter anywhere else in the world. We feel that the analysis provided in this manuscript which is the analysis of over 50 clinical strains in Bangladesh collected over a 10 year period, which share a niche with ICP2, is particularly relevant to making conclusions about the selective force exerted by ICP2 over time. In addition, we were able to experimentally confirm our hypothesis that these mutations confer ICP2 resistance.